# Structure and Optimization of Checkpoint Inhibitors

**DOI:** 10.3390/cancers12010038

**Published:** 2019-12-21

**Authors:** Sarah L. Picardo, Jeffrey Doi, Aaron R. Hansen

**Affiliations:** 1Department of Medical Oncology, Princess Margaret Cancer Centre, 700 University Avenue, Toronto, ON M5G 1X6, Canada; aaron.hansen@uhn.ca; 2Department of Pharmacy, Princess Margaret Cancer Centre, 610 University Avenue, Toronto, ON M5G 2M9, Canada; jeffrey.doi@uhn.ca

**Keywords:** checkpoint inhibitors 1, protein structure 2, pharmacokinetics 3, drug optimization 4

## Abstract

With the advent of checkpoint inhibitor treatment for various cancer types, the optimization of drug selection, pharmacokinetics and biomarker assays is an urgent and as yet unresolved dilemma for clinicians, pharmaceutical companies and researchers. Drugs which inhibit cytotoxic T-lymphocyte associated protein-4 (CTLA-4), such as ipilimumab and tremelimumab, programmed cell death protein-1 (PD-1), such as nivolumab and pembrolizumab, and programmed cell death ligand-1 (PD-L1), such as atezolizumab, durvalumab and avelumab, each appear to have varying pharmacokinetics and clinical activity in different cancer types. Each drug differs in terms of dosing, which becomes an issue when drug comparisons are attempted. Here, we examine the various checkpoint inhibitors currently used and in development. We discuss the antibodies and their protein targets, their pharmacokinetics as measured in various tumor types, and their binding affinities to their respective antigens. We also examine the various dosing regimens for these drugs and how they differ. Finally, we examine new developments and methods to optimize delivery and efficacy in the field of checkpoint inhibitors, including non-fucosylation, prodrug formations, bispecific antibodies, and newer small molecule and peptide checkpoint inhibitors.

## 1. Introduction

Checkpoint inhibitors (CPIs) induce an anti-tumor immune response by antagonizing suppressive immune checkpoint regulatory pathways. The recognized function of these immune checkpoints is to modulate or prevent autoimmune responses and or auto-inflammation. The advent of antibodies targeting programmed cell death protein-1 (PD-1), programmed cell death protein ligand-1 (PD-L1) and cytotoxic T-lymphocyte associated protein-4 (CTLA-4) has led to the development of drugs targeting these pathways in the last 10 years. However, their variable pharmacokinetics and response rates has led to efforts to optimize these drugs, as well as to develop new drugs targeting other checkpoint pathways. Here we examine the structure and mechanism of action of these drugs and human pharmacokinetics in terms of their binding affinities, clearance, and the significance of dosing regimens. In addition, we describe efforts to enhance the delivery and formulation of CPIs, while attempting to minimize the immune-related adverse events (irAEs) associated with these treatments.

## 2. CTLA-4, PD-1 and PD-L1 Proteins and Antibodies

### 2.1. Proteins

#### 2.1.1. CTLA-4

CTLA-4 was first described in 1987 as “a new member of the immunoglobulin superfamily” [1]. It is a 223 amino acid protein which is expressed on activated T cells co-expressing CD28 [2] and has extracellular, transmembrane and intracellular components. Its ligands are CD80 (B7-1) and CD86 (B7-2), found on antigen presenting cells and T-regulatory (T-reg) cells, with binding causing downregulation of activated T cell activity and upregulation of suppressive T-reg function. The importance of CTLA-4 is demonstrated in CTLA-4-knockout mice, who develop early and catastrophic immune hyperactivation causing myocarditis and pancreatitis, and die by 3–4 weeks of age [3].

#### 2.1.2. PD-1 and PD-L1

The PD-1 protein is a 288 amino acid protein which is primarily expressed on T cells, but also on other immune cells, such as B cells, natural killer T cells, and monocytes. It was first identified at a gene level in murine cell lines and was initially thought to be involved in apoptosis, as its expression was induced when thymocyte cell death was induced [4]. Subsequently, it was found to suppress immune responses, and, in particular, it is hypothesized that PD-1 suppresses anti-self-responses [5,6]. This theory is supported by the fact that PD-1 induction is suppressed in the presence of “foreign” antigens such as lipopolysaccharide (LPS) and a stimulatory CpG-containing oligodeoxynucleotide CpG1826 [7]. The protein itself has an intracellular domain, a hydrophobic transmembrane domain and an extracellular immunoglobulin domain which is folded into a β-strand “sandwich” connected by a disulphide bridge. The intracellular domain, or cytoplasmic tail, contains an N-terminal sequence which forms an immunoreceptor tyrosine-based inhibition motif, as well as a C-terminal sequence which forms an immunoreceptor tyrosine-based switch motif. The murine and human forms of PD-1 share a 62% identical sequence, but there are significant differences in the ligand-binding sites, including alterations in size, polarity and charge [8].

The PD-1 protein has two major ligands—PD-L1 and PD-L2. Both ligands contain an N-terminal domain, which binds to PD-1, and a C-terminal domain, the function of which is as yet unknown. Both domains have an immunoglobulin-like fold forming a β-strand sandwich similar to that of PD-1 and are joined by a short linker. Nuclear magnetic resonance characterization suggests that PD-L1 proteins form homodimers, exposing the hydrophobic PD-1 binding sites, although whether this occurs in vivo remains unclear [8,9,10]. The PD-L2 molecule has a similar structure, with two immunoglobulin domains and a linker region, with most of the residues in the binding interfaces of both ligands conserved [11].

The binding of human PD-1 and PD-L1 proteins forms a 1:1 complex and induces a conformational change in PD-1, with the closure of the CC’ loop around PD-L1 and formation of hydrogen bonds, which are hypothesized to stabilize the complex and cause re-arrangements of the PD-1 protein [10,12]. The binding regions contain both hydrophobic and polar sites, with the majority of the interaction occurring in the front strands of both proteins using the large hydrophobic surfaces of the immunoglobulin-V-type domains; the complex between PD-1 and PD-L2 is thought to be similar, although much of this work is only in murine proteins [11].

#### 2.1.3. Significance in Cancer Immunity

CTLA-4 was the first checkpoint molecule targeted in cancer treatment, initially in melanoma with dramatic results, and subsequently in other cancer types. Its significance in anti-tumor immunity was described over 20 years ago in murine models where blockade of CTLA-4 caused tumor rejection both in established tumors and with secondary exposure to tumor cells [13]. PD-1 is mainly expressed on immune cells, in particular T lymphocytes, as well as B lymphocytes, NK cells, dendritic cells and monocytes, and its expression can be induced by many factors, including interleukins, infectious agents and LPS [14,15,16]. As described above, its main function is in immune suppression; therefore, in tumors, it can have the detrimental effect of decreasing anti-tumor immunity, particularly because many cancers develop the capability to express the PD-L1 ligand. On presentation of an antigen to a T lymphocyte, a typical T-cell response involves binding the antigen to the specific T-cell receptor, expansion of this T cell clone and, finally, an effector phase of the response. The co-receptors CD28 and CD3 are involved in the induction of this response. Specifically, in the tumor microenvironment, neoantigens from cancer cells are released, captured and processed by antigen-presenting cells. Antigen presentation to T cells must be accompanied by a secondary signal mechanism in order for T cells to be primed and activated. This secondary signal can be via cytokines, such as IL-12 and type 1 interferon, factors released by dying cancer cells or via the gut microbiota [17,18]. Both CTLA-4 and PD-1 suppress CD28-mediated pathways; PD-1 does this by the activation of phosphatidylinositol-3-kinase which in turn inhibits Akt phosphorylation, thereby suppressing T-cell activation, and also inhibits glycolytic pathways, thereby decreasing cellular metabolism [19]. CTLA-4 binds to its B7 ligands with a much higher affinity than CD28, preventing T-cell stimulation.

Tumor cells in many cancer types express PD-L1 and therefore can activate this pathway to escape immune surveillance. The expression of PD-L1 by tumor cells may be an adaptive response to anti-tumor immune response, with PD-L1 expression co-localized with tumor-infiltrating lymphocytes and IFN-δ, an inflammatory cytokine [20]. However, the clinical significance of PD-L1 expression is tumor histology-specific, with some cancers demonstrating improved outcomes with high PD-L1 expression, while, in other tumors, PD-L1 expression does not correlate with better survival [21,22,23,24,25,26]. The expression of PD-1 and PD-L1 in tumors may also be heterogeneous both intra-tumorally and between primary and metastatic tumor sites [27,28,29,30].

### 2.2. Monoclonal Antibodies

#### 2.2.1. Anti-CTLA-4

Ipilimumab, which binds to CTLA-4, was the first CPI to be licensed in 2011, and was initially used for the treatment of metastatic melanoma but is now indicated in multiple tumor types. It has a high surface area at its binding site and has a dissociation constant of 5.25 nM, with a large surface area buried at its binding surface with CTLA-4 [31] (Table 1). Tremelimumab is another monoclonal antibody targeting CTLA-4 but has not yet been licensed for any indication, although it has orphan drug status for treatment of mesothelioma. Tremelimumab is an IgG2 antibody; this subtype is thought to have less complement activation and antibody-dependent cell-mediated cytotoxicity [32]. It is currently in ongoing clinical trials, in particular in combination with durvalumab [33].

#### 2.2.2. Anti-PD-1/PD-L1

The first two anti-PD-1 CPIs licensed were nivolumab and pembrolizumab, based on their anti-tumor activity in phase I studies [34,35,36]. Pembrolizumab is an IgG4 human antibody; these antibodies have a low affinity for C1q and Fc receptors compared to other IgG molecules, making them a good antibody choice for immunotherapy, with the lowest chance of host immunity stimulation [37]. Most IgG4 antibodies are capable of a process called Fab arm exchange, in which half-molecules (a heavy chain and attached light chain) can be exchanged between IgG4 molecules [38]; pembrolizumab has a hinge region containing a S288P mutation, which prevents Fab arm exchange due to a conformational change [39,40]. The structure of nivolumab is very similar; it is an IgG4 antibody which differs from pembrolizumab only in the variable region of epitope binding-pembrolizumab binds to the C’D loop and nivolumab binds to the N-terminal loop on the PD-1 molecule [41].

Atezolizumab was the first anti-PD-L1 antibody licensed in the US. Atezolizumab and the other licensed anti-PD-L1 antibodies avelumab and durvalumab are IgG1 antibodies, which bind to the front beta-sheet of PD-L1. The heavy chain and light chain regions of these antibodies are involved in binding, with varying buried surface areas on each molecule which may affect their binding affinities [42,43]. These three antibodies have been noted to use all three complementarity determining regions from their heavy chains and two from the light chains [43,44].

After ipilimumab was licensed for the treatment of metastatic melanoma in 2011, the anti-PD-1 and anti-PD-L1 CPIs were subsequently approved for the treatment of many other cancer types, in the metastatic, adjuvant and neo-adjuvant settings. Initial approvals were for refractory/advanced melanoma and non-small cell lung cancer (NSCLC) for the anti-PD-1 CPIs, with subsequent licensing for their use in head and neck cancers, renal cell carcinoma, Hodgkin lymphoma and urothelial carcinomas [45]. Interestingly, the anti-PD-1 antibody pembrolizumab was the first oncologic therapy to be approved for use on the basis of a genetic alteration, with FDA approval granted in 2017 for its use in any tumor demonstrating microsatellite instability (MSI) [46]. The anti-PD-L1 antibodies are used in urothelial, kidney, lung and Merkel cell carcinoma, with many further studies ongoing. The presence of high tumor mutational burden (TMB) (the number of somatic tumor mutations per megabase of sequenced DNA) may identify tumors that are more likely to respond to CPI, such as those tumors that are microsatellite-unstable; however, to date, high TMB is not used to select therapy for patients [47]. Interestingly, responses to CPIs can be durable, with subsets of patients achieving long-lasting complete responses in some disease types, although, for many others, immune escape mechanisms develop, allowing tumors to evade the response primed by CPIs [48]. These treatments generally have a high tolerability, although the main toxicities, which are immune-related inflammatory effects, may be serious in a subset of patients.

#### 2.2.3. Binding Affinities and Pharmacokinetics

Nivolumab has a binding affinity to the PD-1 protein of 3.06 nM, while pembrolizumab has an even higher affinity, with a dissociation constant of 27 pM, possibly due to its extensive binding sites to PD-1, which include hydrogen bonds, specifically water-mediated hydrogen bonds, and salt bridges [41,49,50]. Interestingly, pembrolizumab has a much lower affinity for mouse PD-1, which may be explained by specific amino acid substitutions (Asp^85^ to Gly^85^) which, when mutated in human PD-1, abolish pembrolizumab binding. Atezolizumab has a high binding affinity of 0.4 nM, utilizing specific hot-spot residues on the protein binding surface [42,51], while avelumab and durvalumab have dissociation constants of 42.1 pM [43] and 667 pM [52], respectively.

Studies have shown moderate inter-individual variability (IIV) in pharmacokinetics of CPIs. Ipilimumab has stable clearance over dose ranges from 0.3 to 10 mg/kg, with a half-life of 14.7 days and IIV largely influenced by body weight and baseline LDH value, while age, gender, renal and hepatic function do not affect clearance [53]. The steady state trough concentration of ipilimumab is a predictor of response, with higher trough concentrations (in patients receiving higher doses) resulting in improved complete response rates and higher overall survival (OS), but also in increased rates of irAEs [54,55]. Both the anti-PD-1 antibodies nivolumab and pembrolizumab have linear clearance over dose ranges of 0.1–20 mg/kg and 1–10 mg/kg respectively, with both demonstrating a time-dependent decline in clearance rates, although the decline did not appear to impact clinical outcomes [56,57,58]. For the anti-PD-L1 antibodies atezolizumab, avelumab and durvalumab, linear clearance is seen again over wide ranges of doses. For atezolizumab, which is usually used at a fixed dose of 1200 mg, clearance was stable at doses between 1–20 mg/kg and rates were affected by body weight and serum albumin [59]. Avelumab has a similar linear clearance, but interestingly, time-dependent clearance changes differed between tumor types, with Merkel cell carcinoma and head and neck squamous cell carcinoma patients having clearance declines of 24–32%, while all other tumor types had minimal decline in clearance over time [60]. Durvalumab had linear clearance at doses higher than 3 mg/kg, with numerous factors influencing clearance including albumin, body weight, cancer type and gender [61]. Interestingly, a factor that influences clearance in all three anti-PD-L1 antibodies is the development of anti-drug antibodies, which develop in 31.7%, 4.16% and 3.1% respectively for atezolizumab, avelumab and durvalumab, but are unlikely to be clinically relevant as they did not affect clearance to a meaningful degree.

The antitumor effect of pembrolizumab is driven by the reactivation of adaptive immune response by inhibiting PD-1 expressed on T-cells. Once the PD-1 on T-cells are fully saturated by pembrolizumab, the shape of the exposure–response relationship within the dose range of 2–10 mg/kg or 200 mg (exposure at 2 mg/kg every three weeks is similar to exposure at 200 mg every three weeks) is flat, as demonstrated in multiple indications [62]. Available pharmacokinetics (PK) results in participants with various indications (melanoma, NSCLC, HNSCC, and MSI-H) supporting a lack of meaningful difference in PK among tumor types. Therefore, the selection of the 200 mg every three weeks dosing for pembrolizumab was supported as an appropriate dose for multiple tumor types.

Similarly, nivolumab, dosed at a fixed dose of either 240 mg every two weeks or 480 mg every four weeks results in a similar time-averaged steady state exposure and safety as 3 mg/kg every two weeks across multiple tumor types in numerous clinical trials, and is approved at a fixed dosing for most indications [63,64,65]. Peripheral PD-1 receptor occupancy is saturated at doses ≥ 0.3 mg/kg after eight weeks treatment, again supporting minimizing the doses administered, although the degree of intra-tumoral receptor occupancy is not yet known [66]. Some regulatory authorities have suggested weight-based dosing for patients less than 80 kg and fixed dosing above, to avoid unnecessarily high doses for lower-weight patients [67]. Avelumab is currently approved at a weight-based dosing of 10 mg/kg, but simulations suggested that similar risk/benefit profiles would result from fixed dosing at 800 mg, leading to FDA approval of this fixed dose [68]. Issues with cost and drug wastage are also improved with flat dosing [69]; these results are leading to a move towards fixed dosing in many CPI indications and trials, as evidence from the majority of CPIs demonstrates that exposure, efficacy and safety are similar to weight-based dosing.

#### 2.2.4. Immune-Related Adverse Events

A full discussion of the irAEs associated with CPIs is beyond the scope of this review, but, briefly, these side effects are due to off-target activation or dysregulation of the immune system, which can affect any body organ or system. Common organs affected include the bowel, causing colitis, which can be severe, the lungs, causing pneumonitis, the thyroid gland, which can cause both overproduction or underproduction of the thyroid hormone, the adrenal or pituitary glands, the liver and the skin [70]. There appear to be some patterns to the frequency of irAEs with various CPIs, with colitis and hypophysitis more common with the anti-CTLA-4 antibodies and pneumonitis and hypothyroidism more frequently seen with anti-PD-1 therapies [71]. Deaths from irAEs are rare but do occur, with the most common causes being severe colitis and pneumonitis [71]. Rates of grade 3–4 irAEs increase with combination treatment compared with single agent treatment; for example, treatment of metastatic melanoma with ipilimumab and nivolumab resulted in 59% grade 3–4 AEs, compared with 21% for nivolumab alone and 28% for ipilimumab alone [72]. The management of irAEs includes use of steroids for less severe cases, and immunosuppression for more severe cases, using agents such as infliximab and mycophenolate [73].

## 3. Optimization of Checkpoint Inhibitors

While CPIs are part of standard of care in multiple tumor types, efforts to optimize these antibodies to improve their efficacy and safety are currently underway.

### 3.1. Non-Fucosylated Antibodies

Non-fucosylated antibodies have been modified so that the glycans in the Fc binding portion of the antibody are not fucose-bound. This modification enhances the antibody-dependent cell-mediated cytotoxicity (ADCC) via the enrichment of Fc-gamma-receptor-expressing effector cells and depletion of T-regulatory cells [74,75,76,77]. A non-fucosylated variant of ipilimumab has been constructed, and, in mice, demonstrated increased anti-tumor activity, peripheral T-cell activation and T-reg depletion compared with standard ipilimumab, and also enhanced T-cell-mediated vaccine responses in macaques [76,78]. A modified molecule, similar to atezolizumab but with reduced core fucosylation, demonstrated increased binding to Fc-gamma-receptor-IIIa and enhanced ADCC against PD-L1-expressing tumor cells in a cell-line model [79]. Knockout of the fucosyltransferase gene FUT8 or the pharmacologic inhibition of this gene, which decreased fucosylation, resulted in decreased PD-1 expression and increased T-cell activation in mice, again supporting this as a potential mechanism to enhance the activity of checkpoint inhibitors [80]. Phase I trials of non-fucosylated ipilimumab are enrolling.

### 3.2. Pro-Drug Formulations

Prodrug formulations of antibodies utilize a masking peptide that binds to the antigen-binding site of the CPI which reduces systemic activity. When the antibody reaches the tumor site, proteases cleave the masking peptide and the antibody becomes fully functional, allowing tumor-targeted activity and theoretically reducing off-target systemic adverse effects. Prodrug versions of ipilimumab have been developed and demonstrate equivalent anti-tumor and immune activity and reduced lymphohistiocytic inflammation in the gastrointestinal tract and kidneys compared with standard ipilimumab [76,78]. The result is an improved safety profile. Probody^TM^ therapeutics are protease-activated antibodies which have shown pre-clinical efficacy targeting PD-L1 with minimal systemic auto-immunity [81,82]; the Probody drug CX-072 is now in phase I/II clinical trial for solid tumors and lymphoma [NCT03013491].

### 3.3. Bispecific Antibodies

Another method to optimize CPIs is to fuse them to another antibody which can then simultaneously bind another target molecule. These molecules then have the extracellular domains of two separate antibodies, both of which can bind to their respective ligands and retain their signaling activity. An example of this type of protein is the PD1-Fc-OX40L molecule, which, on testing, retained its high affinity binding for both PD-L1/L2 and OX40, caused T-cell activation and also demonstrated an improved anti-tumor immune response compared with single antibody treatment or the combination of the two separate PD-1 and OX40 antibodies [83]. A bispecific antibody to CTLA-4 and OX40 has also been effective in pre-clinical models, reducing tumor growth and enhancing response to PD-1 targeted therapy, and is now in phase I clinical trials [NCT03782467] [84]. The RANK/RANKL pathway is usually associated with bone homeostasis and is targeted using bone-protective agents, such as denosumab in patients with metastatic bony lesions and with osteoporosis [85]. However, this pathway is also involved in the tumor-associated immune response, with increased RANKL expression seen in tumor-infiltrating T-cells and RANK expression on dendritic cells and immunosuppressive M2 macrophages [86]. While trials are underway combining CPIs with denosumab, bispecific antibodies targeting the PD-1/PD-L1 and RANK/RANKL pathways have been developed, and show significant anti-tumor activity in mouse models, in particular those of colon and lung cancer [87]. This activity was dependent on CD8+ T cells and IFN-ɣ, and could be increased further by combining the bispecific antibody with an anti-CTLA4 antibody.

Bispecific antibodies have already entered early phase clinical trials. A fusion protein consisting of an anti-PD-L1 antibody fused to the extracellular domain of TGF-β receptor II, M7824, showed excellent pre-clinical activity, suppressing metastases, inducing long-term anti-tumor immunity and improving OS in mouse models of breast and colon cancer, both as a single agent and in combination with a therapeutic cancer vaccine [88,89]. It is currently in phase I/II trials in many cancer types including breast, prostate, lung, biliary tract and colorectal, with an early biliary tract cancer trial showing an overall response rate of 27% [PMC6421177, PMC6421170]. Another bispecific antibody, MGD013, which targets PD-L1 and LAG-3, another CPI, has shown pre-clinical activity and is in phase I trials in solid tumors [NCT03219268] [90,91]. Issues that arise with bispecific antibodies include the potential for increased immunogenicity and therefore more adverse events, as well as difficulties with safety assessments in animal models. There are many other bispecific antibodies in pre-clinical development, combining immune checkpoint blockade with other tumor-specific protein binding.

## 4. New Agents Targeting Immune Checkpoints

### 4.1. Small Molecule Checkpoint Inhibitors

While there has been considerable progress in the development of antibodies targeting the PD-1/PD-L1 pathway, interest has been growing in attempts to block this axis using small molecules. The purported benefits of using small molecules rather than antibodies include potentially better oral bioavailability, fewer immune-related adverse events, improved tumor penetration and a lower production cost. The initial molecules shown to inhibit this pathway were sulfamonomethoxine and sulfamethizole, which could rescue PD-1-mediated inhibition of IFN-g production, a process which was dependent on PD-L2 [92]. Substituting particular rings in the structure of the sulfamethizole compound, such as a phenyl ring instead of a pyridyl ring, improved the efficacy of the compound in restoring IFN-ɣ expression. While, ultimately, research into these compounds was not continued, they provided proof of concept for the small molecule inhibition of the PD-1/PD-L1 pathway.

Several other small molecule compounds that inhibit PD-L1 have been patented [93]. These molecules have been shown to bind directly to each dimer of PD-L1 and can dissociate the PD-1/PD-L1 complex, and certain “hot spots” on the PD-L1 molecule, which are targetable by small molecules, have been identified using in vitro studies of these compounds [94,95]. However, one of the major problems with small molecule inhibitors to date has been their large molecular weight, which impairs adequate absorption and distribution in the human body.

The only small molecule currently in human clinical trials is a molecule called Ca-170, which inhibits both the PD-L1 pathway and the V-domain Ig suppressor of the T-cell activation (VISTA) pathway. Pre-clinical work has demonstrated that in mice, this molecule can inhibit tumor growth, enhance peripheral T cell activation and increase activation of tumor-infiltrating CD8+ T-cells [96,97]. Oral bioavailability in mice was 40%, but in monkeys was <10%, again raising the issue of oral administration of these compounds. Ca-170 is in phase 1 clinical trials in patients with advanced solid tumors and lymphoma, and also in phase II trials, with a clinical benefit rate of 59.5% reported, and higher response rates seen at lower doses [98]. Interestingly, a recent study examining the mechanism of binding of Ca-170 has shown that there is no direct binding between the compound and the PD-L1 molecule, suggesting there may be an alternative mechanism of action [99]. To date, the majority of small molecule inhibitors of PD-L1 do not appear to be ready for widespread clinical usage and further pre-clinical work is needed to optimize their formulation and use.

### 4.2. Peptide Checkpoint Inhibitors

As described above, the crystal structure of the PD-1 and PD-L1 molecules and the mechanism by which they bind has been clearly defined, and, therefore, interest has grown in designing a peptide inhibitor that could bind to one of these binding sites. With this data, the first peptide antagonist, (D)PPA-1, was described in 2015, and designed using a mirror-image phage display method, binding to PD-L1 and blocking the PD-1/PD-L1 interaction and decreasing tumor growth in vivo [100]. Replacing the L-amino acids with D-amino acids can improve the stability and oral bioavailability of these drugs. Another more recently developed peptide, PL120131, was designed to interact with the PD-1 molecule, based on the interacting residues on PD-L1 from the amino acid glycine at position 120 to asparagine at position 131 [101]. PL120131 was shown to act as a competitive inhibitor of PD-L1 by associating with the binding groove on PD-1, and to reverse the apoptotic signal induced by soluble PD-L1 in Jurkat cells and primary lymphocytes. Another class of peptides are the macrocyclic peptides, which bind to the PD-1-binding site on the PD-L1 molecule, and can restore T-cell function in vitro [102].

To date, none of the peptide inhibitors of the PD-1/PD-L1 pathway have been used in human trials. The peptide molecule TPP-1 has a high affinity for human PD-L1, and, in a mouse model, could decrease tumor growth by 56% compared with control peptide-treated mice, by re-activating T cells through blocking the PD-1/PD-L1 interaction [103]. A compound called UNP-12 demonstrated a 44% reduction in tumor growth in mice [104,105]. More recently, NP-12, which also inhibits the PD-1/PD-L1 interaction and can inhibit tumor growth and metastases in colon and melanoma mouse models, demonstrated improved efficacy when combined with tumor vaccination or cyclophosphamide [106]. The peptide inhibitors are still in early phases of development but may provide an alternative method through which to inhibit immune checkpoints.

## 5. Conclusions

CPIs have changed the landscape of cancer treatment in recent years, with a small proportion of patients with a variety of tumors experiencing deep and durable responses. Understanding the pharmacokinetics of many CPIs has led to a switch from weight-based to fixed dosing, which is likely to continue as more studies of the efficacy and PK of fixed dosing are completed. IrAEs and heterogeneity in responses has led to efforts to optimize existing CPIs and to develop new methods by which to inhibit checkpoint molecules. Understanding the structure of CPIs and their ligands can help in the further enhancement of these therapeutic agents.

## Figures and Tables

**Table 1 cancers-12-00038-t001:** Checkpoint inhibitors, their pharmacokinetic and dosing profiles and indications.

Agent	Type	Antigen	Clearance	Dissociation Constant/Binding Affinity	Half-Life	Indications	Companion/Complementary Diagnostic Assay	Dosing	Year First Licensed	Pharmaceutical Company
Ipilimumab	IgG1 human antibody	CTLA-4	Stable clearance over doses 0.3–10 mg/kg	Dissociation constant 5.25 nM	15.4 days	Melanoma, renal cell carcinoma, MSI-high colorectal carcinoma	None	Weight-based dosing (1–10 mg/kg)	2011	Bristol Myers Squibb
Tremelimumab	IgG2 human antibody	CTLA-4	Stable clearance over doses 10–15 mg/kg	Binding affinity 0.28 nM	22 days	None as yet	None	Weight-based dosing (3–15 mg/kg) or fixed dosing (75 mg)	Not yet licensed	AstraZeneca
Nivolumab	IgG4 human antibody	PD-1	Linear clearance over doses of 0.1–20 mg/kg	Dissociation constant 1.45 nM	25 days	Melanoma, non-small cell lung cancer, renal cell carcinoma, small cell lung cancer, head and neck cancer, hepatocellular carcinoma, Hodgkin lymphoma, urothelial cancer, MSI-high or mismatch repair-deficient colorectal cancer	Dako 28.8 Pharm.Dx assay (complementary)	Weight-based dosing (1–3 mg/kg) or flat dosing (240 mg)	2014	Bristol Myers Squibb
Pembrolizumab	IgG4 human antibody	PD-1	Linear clearance over doses 1–10 mg/kg	Dissociation constant 29 pM	22 days	Melanoma, non-small cell lung cancer, renal cell carcinoma, small cell lung cancer, Hodgkin lymphoma, primary mediastinal large B-cell lymphoma, Merkel cell carcinoma, hepatocellular carcinoma, gastric cancer, renal cell carcinoma, endometrial carcinoma, cervical cancer, head and neck cancers, urothelial carcinoma, gastric/GEJ/esophageal cancers, mismatch repair deficient tumors	Dako 22C3 Pharm.Dx (companion for non-small cell lung cancer, gastric or gastroesophageal junction adenocarcinoma, cervical cancer, urothelial carcinoma, head and neck squamous cell carcinoma, and esophageal squamous cell carcinoma)	Fixed dosing (200 mg)	2014	Merck
Atezolizumab	IgG1 human antibody	PD-L1	Linear clearance over doses 1–20 mg/kg	Binding affinity 971 Å2	27 days	Urothelial carcinoma, non-small cell lung cancer, triple-negative breast cancer, small cell lung cancer	Ventana SP142 (companion for urothelial carcinoma and triple-negative breast carcinoma)	Fixed dosing (840 mg, 1200 mg, 1680 mg)	2016	Genentech
Avelumab	IgG1 human antibody	PD-L1	Linear clearance over doses 1–20 mg/kg	Binding affinity 875.4 Å2	6 days	Merkel cell carcinoma, urothelial carcinoma, renal cell carcinoma	None	Fixed dosing (800 mg) or weight-based dosing (10 mg/kg) (not Food and Drug Administration (FDA) approved)	2017	EMD Serono/Pfizer
Durvalumab	IgG1 human antibody	PD-L1	Linear clearance at doses higher than 3 mg/kg	Dissociation constant 667 pM	18 days	Urothelial carcinoma, non-small cell lung cancer	Ventana SP263 (complementary)	Weight-based dosing (10 mg/kg) or fixed dosing (1500 mg) (not FDA-approved)	2017	AstraZeneca

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
