# Peer review of "Structure and Optimization of Checkpoint Inhibitors"

_cancers, 2019, doi:10.3390/cancers12010038_

Round 1

Reviewer 1 Report

In this manuscript, by reviewing recent publications, the authors provide an information about structure of CTLA-4, PD-1 and PD-L1 and anti- CTLA-4, PD-1 and PD-L1 monoclonal therapeutic antibodies including binding affinities and pharmacokinetics. Moreover, authors provide an information about recent optimization and new therapeutic approaches targeting CTLA-4, PD-1/PD-L1 and other checkpoint molecules.

It is very interesting and an attractive review for both clinicians and basic researchers. The manuscript is well written and only minor revision are requested.

The use of abbreviations in text can be improved. Some abbreviations are introduced but not used later or used only sometimes, for instance NSCLC in text and table1. Abbreviation for immune-related adverse events (irAEs) is introduced but not used later. Abbreviation for Microsatellite instability should be introduced MSI as later “MSI” is used. Abbreviation HNSCC should be introduced. Line 185: It might be easier for readers to write directly “every 3 weeks” instead “Q3W” as in following paragraph, “every 2 weeks” is used, not Q2W. Please, use always the same brackets [XXX] for clinical trials numbers – line 258 (NCT03219268). Line 101-104: “However, the clinical significance of PD-L1 expression is tumor histology specific, with some cancers demonstrating improved outcomes with high PD-L1 expression, while in other tumors PD-L1 expression does not correlate with better survival.” References can be enriched by literature from other type of cancer : Melanoma - DOI: 1200/JCO.2016.67.2477 ; Hepatocellular carcinoma  doi: 10.3390/cancers11101554

Author Response

Thank you for the review and helpful comments and suggestions Abbreviations have been modified as per the reviewer's suggestions Line 185 has been modified as per the reviewer's suggestions Clinical trial numbers have been modified to use consistent brackets References have been added as per the reviewer's suggestions

Reviewer 2 Report

The manuscript entitled "Structure and optimization of checkpoint inhibitors" by Picardo et al inherited about a systemic revision of landscape immunotherapeutic drugs focusing on pharmacokinetics features of each molecule is well written and suitable for pubblication after minor revisions:

in the introduction section, please, could the authors offer a more detailed description of pharmacokinetics and pharmacodynamics feature of each molecular drug envolved in this review? in the section introduction, please could the authors report the adversae events (AE) relative to immunotherapeutic regimen? please, could the authors remove from table 1 pharmaceutical company lane? please, could the authors produce a image comphrensive of molecular mechanism described  in the paper?

Author Response

Many thanks for your review and comments and suggestions The introduction has been modified to include the suggestions of the reviewer We feel that the pharmaceutical company who market the drugs is a useful piece of information relating to the use of the drug and for drug combinations We feel that an image relating to the mechanism of action of these drugs is unnecessary as it is not the focus of this paper and there are many reviews already published describing the mechanism of action in great detail.